# Kinetics of Manganese Peroxidase Using Simple Phenolic Compounds as Substrates

**DOI:** 10.3390/metabo15040254

**Published:** 2025-04-09

**Authors:** Madeline G. Gruenberg, Jonathan J. Halvorson, Michael A. Schmidt

**Affiliations:** 1Department of Biochemistry and Molecular Biology, Wright State University, Dayton, OH 45435, USA; gruenberg.3@wright.edu; 2Northern Great Plains Research Laboratory, United States Department of Agriculture, Agricultural Research Service, Mandan, ND 58554, USA; jonathan.halvorson@usda.gov

**Keywords:** manganese peroxidase, tannins, soil organic matter, kinetics, phenolic compounds

## Abstract

**Background/Objectives:** Secondary metabolites encompass diverse groups of compounds; one such group is phenolics, which include small phenols up to larger polyphenols such as lignin and tannins. Smaller compounds such as phenolic acids can serve as substrates for soil microbes and enzymes. The specific interaction between plant secondary metabolites (PSMs) and soil enzymes determines whether the products of these reactions contribute to the formation of soil organic matter (SOM) or are degraded into small organic molecules. Methods: Here, we monitored the activity of a redox active soil enzyme, manganese peroxidase (MnP), with three small phenolic compounds. The compounds used in this study were pyrogallol, gallic acid, and benzoic acid. **Results:** Based on the kinetic parameters determined, pyrogallol and gallic acid are both substrates for MnP with different products and kinetics. **Conclusion:** Pyrogallol reacts faster and produces a more stable quinone than gallic acid. Benzoic acid is not a substrate for MnP.

## 1. Introduction

PSMs are an integral component of several critical processes in soil including nutrient cycling, detoxification, and redox reactions [1]. Secondary metabolites are a diverse class of compounds. One subset of PSMs is phenolics, which encompass compounds ranging from simple phenolic acids to large polymeric structures such as lignin and tannins [2]. Tannins are present in high concentrations in plants, and are the fourth most abundant biomolecule on earth [3]. There are two major classes of tannins: condensed and hydrolyzable [4]. Condensed tannin are polymers of flavan-3-ol units that can be as simple as a dimer, and as complex as 50 subunits. The chemical characteristics of condensed tannins are further varied by the hydroxylation pattern of the B ring. Hydrolyzable tannins contain derivates of gallic acid esterified to a carbohydrate center. The galloyl groups on hydrolyzable tannins can be oxidatively coupled to further modify the functionality of the compound [5,6]. Phenolic acids are produced by all plant types and are classified into two different types: benzoic acid or cinnamic acid derivatives. Benzoic acid derivatives have a carboxylic acid group attached to an aromatic ring. Cinnamic acid derivatives have an unsaturated carboxylic acid group [7]. PSMs are produced by all types of plants with the amount and exact mixture being species- and environmentally dependent [8]. This complexity makes understanding specific PSM–soil interactions difficult, as plant extracts are often mixtures of several different compounds that may have different reactivity in soil.

There are four main pathways through which PSMs enter soil: volatilization and diffusion from plant tissue, the leaching of plant material above the ground, exudation from plant roots, and litter decomposition [9]. Once in the soil, PSMs may adsorb to the soil depending on the polarity of the molecule, with larger, more hydrophobic molecules more likely to partition onto the soil surface [10]. The half-life of secondary metabolites depends on the exact structure, with larger phenolic structures generally more recalcitrant than smaller compounds. As phenolic compounds degrade, they may contribute to soil organic matter (SOM) through both abiotic and biotic processes [11]. The degradation of PSMs tends to be slower than that of proteins and carbohydrates; in addition, PSMs affect the decomposition of other compounds that contribute to SOM formation [11]. Within the soil, PSMs also regulate the microbial environment by acting as signals impacting the composition and activity of the microbial community [2].

PSMs affect the formation of SOM by interacting with the soil microbiome, affecting microbiome composition, nutrient usage, and alleviating pathogenic effects [12], with these interactions dependent on environmental stressors [13]. Changes to the environment including temperature, pH, and moisture impact soil microbial communities and the secondary metabolites produced by plants [14]. One possible mechanism of PSMs, especially for simple phenols, through which they can affect the composition of the soil microbiome, is by selecting for organisms that more easily use these compounds as substrates for growth [15]. Previous reports of the respiration of PSMs by soil microbes have been mixed, with the amount of respiration dependent on the exact type of PSMs in the extract being studied [16]. In addition to interacting with microbes, PSMs can also alter the activity of extracellular enzymes such as MnP [17]. Previous reports have suggested that MnP is responsible for most of the degradation of PSMs in soil. In a previous study, Kuan et al. demonstrated that MnP oxidizes lignin to yield a variety of low-molecular-weight phenolic compounds, with Km values ranging from 13 to 39 µM [18].

MnP is a naturally occurring, hydrogen peroxide-dependent soil enzyme produced by fungi that colonize soil and litter [17]; the central ferric heme oxidizes Mn(II) to Mn(III) [19]. It is produced by fungi in the basidiomycetes families, species of mushrooms that include agaricales, corticiales, polyporales, and hymenochaetales [20]. These organisms are broadly distributed in many types of environments [21]. The levels of MnP in soil vary; Merino et al. quantified the levels of MnP in soil samples taken from southern Chile after a five-day incubation in Koroljova–Skorobogat’ko medium at pH 5. The levels of MnP ranged from 6 to 75 μg g^−1^ [22]. The role of MnP in oxidizing large-molecular-weight compounds such as lignin has been investigated [23], as well as the role of smaller acids in the reaction [24,25]. Although the interaction with MnP and large molecular compounds has been documented, the rate that MnP oxidizes different phenolic compounds has not been investigated.

Determining the specific effect of tannins on soil can be difficult. In addition to analytical difficulties of using large complex compounds, many researchers use plant extracts that are often a mixture of several different compounds, leading to varying results [26,27]. Even commercially available forms of a commonly used hydrolyzable tannin, tannic acid, are mixtures that can vary by manufacturer and lot number [28]. In our previous work, we used the model compounds gallic acid, pyrogallol, and benzoic acid to demonstrate that the abiotic degradation of small phenolic compounds by Mn is dependent on the structure of the phenolic compound and the oxidation state of Mn [29,30]. Gallic acid was more oxidized, producing small organic compounds and CO_2_ when reacted with Mn(IV), while pyrogallol produced a quinone and less CO_2_ under the same conditions. Finally, benzoic acid did not react with Mn(IV). Our hypothesis was that the biotic degradation of small phenols would follow this pattern of degradation, with the carboxylic acid containing gallic acid producing less quinone than pyrogallol. To test our hypothesis, we used a commercially available form of MnP and three model phenol compounds. The model phenolic compounds we chose for this study were gallic acid, pyrogallol, and benzoic acid. We thoughtfully picked these three compounds not only for their own presence in plants and soils, but as model compounds for more complex PSMs such as tannins. Gallic acid and pyrogallol share the same aromatic and hydroxyl functionality; however, gallic acid has an additional carboxylic acid attached to the aromatic ring. Benzoic acid lacks the hydroxyl groups while having the carboxylic acid group of gallic acid (Figure 1). Using these model compounds allowed us to assess how the carboxylic acid and hydroxyl groups affect the interaction with MnP. Using these three compounds as substrates for MnP at 25 °C, we monitored for the formation of a quinone using UV/Vis spectroscopy.

## 2. Materials and Methods

### 2.1. Chemicals

MnP from *Phanerochaet chrysoporium* was purchased from Sigma Aldrich (St. Louis, MO, USA). The enzyme is recombinantly expressed in corn and purchased as a powder with greater than 200 units per gram. Gallic acid was purchased from Acros Organics (Geel, Beligium). Pyrogallol and benzoic acid were purchased from Alfa Aesar (Haverhill, MA, USA). Manganese(II) sulfate monohydrate, sodium acetate, and glacial acetic acid were purchased from ThermoFisher Scientific (Pittsburgh, PA, USA). A stock solution of 0.1 M sodium acetate buffer was produced and the pH was adjusted to 4.5 using acetic acid. This buffer was used for all solutions. All substrates were 98% or more pure according to the manufacturer and purity was previously confirmed using high-performance liquid chromatography [29]. 

### 2.2. Enzyme Reaction Rate

Optimal reaction conditions for other PSMs and MnP have been previously determined and have been modified here for the compounds investigated [17]. Enzyme reactions were monitored using 50 µM MnP (manganese peroxidase) in 0.1 M sodium acetate buffer (pH 4.5) and 0.05% hydrogen peroxide. To determine the length of the reaction, MnP was incubated at 25 °C with 10 mM of each substrate and 0.1 mM Mn(II) separately for 30 min. The absorbance was measured at 400 nm at various time points during the reaction, as indicated in Figure 2. The absorbance was monitored at 400 nm using a ThermoFisher Genesys 50 UV/Vis spectrophotometer. All reactions were performed in triplicate.

### 2.3. Enzyme Kinetics

To determine the kinetics of the reaction of MnP with the phenolic compounds as substrates, the reactions were incubated at 25 °C for 8 min. The reactions were performed with 50 µM MnP in 0.1 M sodium acetate buffer (pH 4.5) and 0.05% hydrogen peroxide. Substrate concentrations for the phenolic compounds ranged from 0.028 to 10 mM. The reactions were performed in the presence and absence of 1 mM Mn(II). The absorbance was monitored at 400 nm using a ThermoFisher Genesys 50 UV/Vis spectrophotometer. All reactions were performed in triplicate.

### 2.4. Data Analysis

Graph Pad Prism software version 10.2.0 (392) was used to determine the kinetic parameters (Km and Vmax) using a least squares regression of the data. A sum of squares F-test was used to determine if the best-fit values of each parameter (Km and Vmax) were statistically different. Quinone formation was calculated using the Beer–Lambert equation with the molar extinction coefficient 1200 M^−1^ cm^−1^ [31].

## 3. Results

### 3.1. Enzyme Reaction Rate

Figure 2 shows the initial reaction rates of MnP with 10 mM of each of the phenolic compounds. Quinone formation reached completion within 20 min for both pyrogallol and gallic acid, while benzoic acid did not serve as a substrate for MnP. To ensure kinetic measurements were capturing the initial rate of the reaction, 8 min was used as the incubation time when determining the kinetic parameters.

### 3.2. Enzyme Kinetics

Following the 8 min incubation of MnP with the substrates, both gallic acid and pyrogallol produced quinone, with pyrogallol serving as the better substrate (Figure 3, Table 1). The reactions involving Mn(II) led to increased quinone formation for both pyrogallol and gallic acid compared to the reactions without Mn(II). The Km for pyrogallol was 0.28 mM compared to 0.66 mM for gallic acid (Table 1). The Vmax followed the same pattern with 0.59 mM min^−1^ for pyrogallol and 0.35 mM min^−1^ for gallic acid. Benzoic acid was not a substrate for MnP. The Km (*p* = 0.03) and Vmax (*p* < 0.001) values for gallic acid and pyrogallol were statically different using the sum of squares F test.

## 4. Discussion

The biotic degradation of polyphenols can be facilitated by various enzymes including laccases, peroxidases, and tyrosinases [11]. Phenol degradation has become an area of interest as researchers aim to better understand the soil environment and the subsequent impact of the reaction products [32]. The oxidation of larger polyphenolic compounds contributes to soil organic matter (SOM), while the oxidation of smaller phenols remains underexplored and is dependent on the phenolic structure. To investigate the biotic degradation of simple phenols in the soil, we used a broadly distributed soil enzyme known to degrade similar structures, MnP [33].

Lignin degradation is crucial for carbon and nutrient cycling [34], and MnP has been well studied for its role in the degradation of lignin [35]. In addition, MnP aids in the oxidation of larger phenolic compounds, whose breakdown contributes to SOM formation [11]. However, the influence of different functional groups on MnP activity, as well as the corresponding enzymatic kinetics, remains unclear. Our study aimed to explore simple phenolic compounds and investigate how different functional groups influence which PSMs are good substrates for MnP. We focused on simple phenolic compounds as model substrates for larger polyphenols, examining how various functional groups influence redox reactions in the soil. The phenolic compounds selected for this study—pyrogallol, gallic acid, and benzoic acid—are simple phenolic structures with different combinations of functional groups that mimic those found in larger polyphenolic compounds such as condensed and hydrolyzable tannins. As stated above, these compounds are also of interest due to their broad distribution in many environments. While all three are simple phenolic compounds, they differ in structure (Figure 1). Using these model compounds, we aimed to determine which functional groups facilitate interaction with MnP.

Each phenolic compound was incubated with and without Mn(II) at 25 °C for 8 min and monitored at 400 nm for quinone formation. In our previous work, we demonstrated that the abiotic degradation of pyrogallol via Mn(IV) resulted in the formation of a quinone [29]. Here, we monitor the kinetics of quinone formation by MnP. The kinetic parameters (Table 1) demonstrate that the reactions with Mn(II) were more efficient than those without Mn, exhibiting a higher affinity (lower Km) and higher Vmax for pyrogallol than gallic acid. Benzoic acid did not show any measurable reaction under either condition. Pyrogallol and gallic acid are both substrates for MnP, but pyrogallol is the better substrate as it yields higher quinone formation and more favorable kinetics. Gallic acid, while still a substrate due to the addition of the carboxylic acid functional group, produced less quinone. In our previous study, we determined by NMR that in the abiotic degradation of gallic acid by Mn(IV), the product was not a quinone [29]; however, a small amount of quinone was produced via biotic degradation via MnP. This demonstrates that the hydroxyl groups on the ring structure are the driving force of the degradation of these phenols to a stable intermediate in soil.

Previously, we observed through the abiotic degradation of the same phenolic compounds that gallic acid had a slower reaction; however, it was fully oxidized to CO_2_, while pyrogallol reacted faster and produced a more stable quinone and less CO_2_ [29]. This pattern of abiotic degradation is consistent with our current findings of biotic degradation via MnP and supports our initial hypothesis. This suggests that the biotic and abiotic degradation pathways are similar and PSMs with free carboxylic acid functionality are more likely to be fully oxidized to CO_2_, and phenolic compounds lacking the carboxylic acid functionality form other degradation products that contribute to SOM. Based on these findings and the observed structural differences, we conclude that both pyrogallol and gallic acid are substrates for MnP, with pyrogallol exhibiting more favorable kinetics for the formation of a quinone. This difference may be attributed to steric hindrance caused by the free carboxylic acid group in gallic acid. This suggests that hydrolyzable tannins with the carboxylic acid group esterified to the carbohydrate are more likely to form a stable quinone and contribute to the formation of SOM. In addition to the role of the carboxylic acid in the reaction, our results suggest that the hydroxyl groups on the phenolic compound are essential for MnP forming a quinone with the phenolic compound as a substrate. Further investigation of the hydroxylation pattern of both hydrolyzable and condensed tannins is necessary to determine how these patterns will influence SOM formation.

## 5. Conclusions

In this study, we have demonstrated that simple phenolic compounds are degraded biotically by a soil enzyme, MnP, and our initial hypothesis that the abiotic degradation and biotic degradation of these simple phenolic compounds would follow the same pattern was supported. Pyrogallol is the better substrate for MnP and produces a quinone that can contribute to the formation of SOM. We also demonstrated that PSMs that have a carboxylic acid functional group are less likely to form a quinone and are not as likely to contribute to the formation of SOM as other PSMs. While this study focuses on small phenolic compounds, it demonstrates how structural differences in PSMs can affect their fate in the environment. Using model compounds that are thoughtfully selected provides a better framework to understand PSM–soil interactions than using crude plant extracts that many contain several different compounds.

## Figures and Tables

**Figure 1 metabolites-15-00254-f001:**
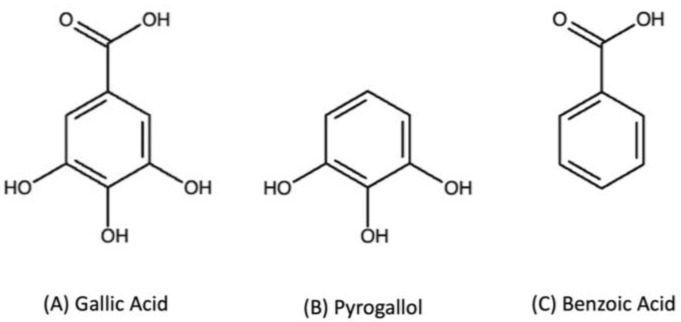
Chemical structures of the compounds used as substrates for MnP.

**Figure 2 metabolites-15-00254-f002:**
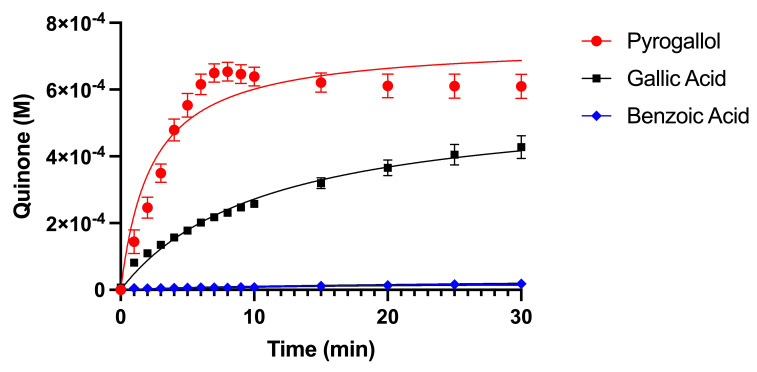
Manganese peroxidase reaction rate with three different substrates. Phenolic compounds at 10 mM were reacted with 50 μM MnP, the temperature was controlled at 25 °C, and reactions were monitored for 30 min. Product formation was measured at 400 nm. All reactions were performed in triplicate with error bars representing standard deviation.

**Figure 3 metabolites-15-00254-f003:**
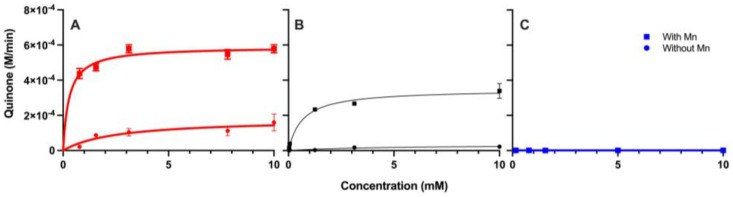
Kinetic plots of MnP with three phenolic compounds. Pyrogallol (red) (**A**), gallic acid (black) (**B**), and benzoic acid (blue) (**C**) at varying concentrations were reacted with 50 μM MnP for 8 min at 25 °C in the presence and absence of Mn(II). Product formation was monitored at 400 nm. The squares indicate the reactions with Mn, and the circles indicate the reactions without Mn. All reactions were performed in triplicate with error bars representing standard deviation.

**Table 1 metabolites-15-00254-t001:** Kinetic parameters of MnP with phenolic compounds.

Compound	Km with Mn (mM)	Km without Mn (mM)	Vmax with Mn (mM/Min)	Vmax without Mn (mM/Min)
Benzoic Acid	N/A	N/A	N/A	N/A
Pyrogallol	0.28	2.6	0.59	0.18
Gallic Acid	0.66	4.3	0.35	0.03

Enzyme reactions were monitored using 50 µM MnP and substrate concentrations ranging from 0.028 to 10 mM. Reactions were incubated at 25 °C for 8 min in the presence and absence of Mn(II). Data were analyzed using Graph Pad software. Kinetic parameters were calculated using Michaelis–Menten kinetics. All reactions were performed in triplicate.

## Data Availability

All data are contained within the manuscript.

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
