# Peer review of "Kinetics of Manganese Peroxidase Using Simple Phenolic Compounds as Substrates"

_metabolites, 2025, doi:10.3390/metabo15040254_

Round 1

Reviewer 1 Report

Comments and Suggestions for Authors

Review Report

Title: Determining the kinetics of manganese peroxidase with simple phenols

Authors: Madeline Grace Gruenberg, Jonathan J Halvorson, Michael Afton Schmidt

Manuscript ID: metabolites-3532004

Type: Article

Journal: Metabolites (ISSN 2218-1989); Section: Plant Metabolism; Special Issue: Advances in Secondary Metabolites: Phytochemical Analysis and Bioactivity Assays

A brief review summary

The aim of the paper was to investigate the kinetic parameters of the enzyme Manganese peroxidase (MnP) with simple phenols Benzoic acid, Gallic acid, and Pyrogallol as substrates, and to compare the enzymatic (biotic) oxidation and degradation of these phenols with the degradation solely catalysed by manganese (abiotic).

While in this study, the kinetic parameters were determined for the reaction with MnP, the abiotic reactions were studied in a previous work from the same group of authors. The element of comparison in the section Discussion is missing.

The paper main contribution is in the description of a model of enzymatic kinetics towards selected simple phenolic compounds.

General concept comments
The main weakness of the presented paper is the small scope of investigation (three simple phenols, one enzyme, restricted to laboratory conditions), the discussion is not enough elaborated and the conclusions are not well-founded. The hypothesis (“the biotic degradation of small phenols follows the same pattern of oxidation as the abiotic degradation”) could be testable, but within the presented paper this was not done. Methodologically, no inaccuracies were detected, only minor remarks which are addressed in specific comments.

Specific comments 

Abstract, Keywords, Introduction

Lines 11-12, Lines 23-25

Secondary metabolites are not a diverse class of compounds, rather: Plant secondary metabolites encompass diverse groups of compounds, among which the group of phenols, from polyphenols such as lignin and tannins to monomeric simple phenols such as benzoic and coumaric acids and their derivatives (Marchiosi et al., 2020; doi:10.1007/s11101-020-09689-2). 

Line 12, Line 19, Lines 25-32

Please explain why you mention tannins in Abstract and Keywords and a great section of the Introduction, when these compounds have not been investigated?

In the introduction section they could be mentioned in relation to the substrates or processes you investigated, but then also you should refer to them in the section Discussion. If the tannins are to be excluded from the manuscript, the references made for them (at least 3 and 4) should also be revised/excluded and the reference list, as well as numbering throughout the manuscript should in that case be revised.

Since MnP is known for oxidation and degradation of lignin, the authors could consider mentioning lignin as well.

Line 38

By „specific PSM-soil“, do the authors mean  „specific PSM-soil interaction“? If so, please add.

Line 40

Shouldn’t instead of “volitization” be “volatilization”?

Line 81

Please specify what exactly is the “pattern of abiotic degradation” from the previous work to which the authors referred, and intended to confirm.

In the discussion section, the results should be interpreted in relation to the “pattern of abiotic degradation”.

Line 91

The formation of the product quinone was monitored, not the “oxidation” of phenols.

Materials and Methods

Line 110

Shouldn’t instead of “varies” be “various”?

Line 124 (similarly, in line 144)

Please rephrase appropriately the sentence “Differences between values was determined using a sum of least squares F test”.

Consider: “statistically significant diffierences” instead of “differences”, “means” instead of “values”, “were” instead of “was”, “sum of squares” instead of “sum of least squares” or just “F-test”.

Which values are being statistically evaluated and what are the results of statistical testing?

Results

Line 152

Please check if the columns in the table are named correctly. It seems that the higher values of Km should be related to the treatment with Mn, instead to the treatment without Mn. Below the table, or in the text, it should be explained how the Km and Vmax were determined, or a brief definition could be added.

Discussion

Lines 169-170, and 171

It can’t be said that substrates’ functional groups affect the enzyme’s activity (nor redox reactions in the soil), but rather that based on differences in functional groups, specific simple phenolic compounds are adequate substrates to MnP, while others are not.

Lines 173-174

Which larger phenolic compounds potential substrates to MnP could that be? Is there any reference for the assumption that a simple phenol as substrate to an enzyme can model for a polymeric phenol as substrate to the same enzyme? In enzymatic catalysed reactions, not only the functional group determines the acceptability as substrate, but also the size of the molecule, and position of functional groups.

Tests with such larger polyphenol molecules could also have been included in these experiments.

Line 193

It can’t be said that “gallic acid was a slower reaction”, please rephrase, e.g. “enzymatic degradation of gallic acid by MnP was found to be slower compared to pyrogallol as substrate”.

Conclusions

Lines 206-207

Abiotic degradation was not part of this investigation, so it can’t be said that “it was demonstrated in this study”. Maybe, if this was better explained at the beginning, to define what is considered biotic degradation, and what abiotic degradation, what is the pattern of abiotic degradation (from previous study) and the one confirmed in this study.

Lines 209-210

It seems that the results are overestimated, or that the conclusions are too generalised (PSM without the carboxylic group) with the assertion that “it was demonstrated that PSM that do not have a carboxylic acid functional group are likely to form a quinone and contribute to the formation of SOM, and it should be removed.

References

Line 240

Publisher Location, Country, page numbering are missing. Please revise the reference according to instructions for authors (https://www.mdpi.com/journal/metabolites/instructions):

For Books and Book Chapters:
2. Author 1, A.; Author 2, B. Book Title, 3rd ed.; Publisher: Publisher Location, Country, Year; pp. 154–196.
3. Author 1, A.; Author 2, B. Title of the chapter. In Book Title, 2nd ed.; Editor 1, A., Editor 2, B., Eds.; Publisher: Publisher Location, Country, Year; Volume 3, pp. 154–196.

The manuscript should be improved: testing of the hypothesis, more focused introduction, more elaborated discussion and conclusions based on results, that is why I suggest major revision.

Kind regards

Author Response

Summary: The authors would like to thank the reviewer for their thoughtful consideration of the manuscript. We believe that edits to the manuscript address the concerns of this reviewer. Below please find the reviewers comments in black and our responses to each comment in red.

Response to comments: 

Comment 1: Secondary metabolites are not a diverse class of compounds, rather: Plant secondary metabolites encompass diverse groups of compounds, among which the group of phenols, from polyphenols such as lignin and tannins to monomeric simple phenols such as benzoic and coumaric acids and their derivatives (Marchiosi et al., 2020; doi:10.1007/s11101-020-09689-2). 

Response 1: Reworded the abstract to reflect this change. Lines 12-13

Reworded the introduction. Line 30

Comment 2: Please explain why you mention tannins in Abstract and Keywords and a great section of the Introduction, when these compounds have not been investigated?

Response 2: Thank you for the thoughtful comment. We are using these small phenols as model compounds for more complex PSM such as tannins. While of interest, the larger compounds are too analytically complex to investigate using our methods. Previously we mentioned in the introduction why these compounds were picked, however we have added some clarifying statements to help the reader understand why tannins are an important concept for this paper. Lines 96-100 have been added with appropriate references.

Comment 3: In the introduction section they could be mentioned in relation to the substrates or processes you investigated, but then also you should refer to them in the section Discussion. If the tannins are to be excluded from the manuscript, the references made for them (at least 3 and 4) should also be revised/excluded and the reference list, as well as numbering throughout the manuscript should in that case be revised.

Response 3: We have revised the introduction and discussion to justify the discussion of tannins. Please refer to the red track changes.

Comment 4: Since MnP is known for oxidation and degradation of lignin, the authors could consider mentioning lignin as well.

Response 4: This is mentioned in lines 80-82.

Comment 5: By „specific PSM-soil“, do the authors mean  „specific PSM-soil interaction“? If so, please add.

Response 5: We have edited the text as suggested. Line 56

Comment 6: Shouldn’t instead of “volitization” be “volatilization”?

Response 6: We have edited the text as suggested. Line 58

Comment 7: Please specify what exactly is the “pattern of abiotic degradation” from the previous work to which the authors referred, and intended to confirm.

Response 7: Thank you for the suggestion. We have added the pattern of abiotic degradation to lines 103-106.

Comment 8: In the discussion section, the results should be interpreted in relation to the “pattern of abiotic degradation”.

Response 8: We have referenced the abiotic degradation in line 268-272.

Comment 9: The formation of the product quinone was monitored, not the “oxidation” of phenols.

Response 9: Thank you for the comment. We have reworded that sentence to reflect that change. Lines 120-121

Comment 10: Shouldn’t instead of “varies” be “various”?

Response 10: Thank you for catching that error. We have changed the word to various, line 142.

Comment 11: Please rephrase appropriately the sentence “Differences between values was determined using a sum of least squares F test”.

Consider: “statistically significant diffierences” instead of “differences”, “means” instead of “values”, “were” instead of “was”, “sum of squares” instead of “sum of least squares” or just “F-test”.

Which values are being statistically evaluated and what are the results of statistical testing?

Response 11: We would like to thank the reviewer for the thoughtful review. We have changed as suggested. Line 176-178.

Comment 12: Please check if the columns in the table are named correctly. It seems that the higher values of Km should be related to the treatment with Mn, instead to the treatment without Mn. Below the table, or in the text, it should be explained how the Km and Vmax were determined, or a brief definition could be added.

Response 12: Thank you again for the careful review of the manuscript.

We have checked the numbers and the numbers for Km are correct in the original document. Km is the substrate concentration to reach ½ Vmax, a lower Km value indicates a higher affinity. A higher Km value indicates a lower affinity. In our study the reactions with Mn should have a lower Km (higher affinity).

However, we did notice in the review that the Vmax for one of the parameters was incorrect by a decimal place. It has been corrected in table 1. The changes does not affect the manuscript.

We have added statements to the table caption to describe how the kinetics are determined.

Comment 13: It can’t be said that substrates’ functional groups affect the enzyme’s activity (nor redox reactions in the soil), but rather that based on differences in functional groups, specific simple phenolic compounds are adequate substrates to MnP, while others are not.

Response 13: This study determined how the functional groups on PSM affect how these compounds are used as a substrate for MnP. We have reworded the sentence to clarify this point. Line 235-236

Comment 14: Which larger phenolic compounds potential substrates to MnP could that be? Is there any reference for the assumption that a simple phenol as substrate to an enzyme can model for a polymeric phenol as substrate to the same enzyme? In enzymatic catalysed reactions, not only the functional group determines the acceptability as substrate, but also the size of the molecule, and position of functional groups.

Response 14: Thank you for the comment. We have added a small statement to clarify that these phenols are intended to mimic larger tannin structures. Lines 248-250.

Comment 15: Tests with such larger polyphenol molecules could also have been included in these experiments.

Response 15: We would like to thank the reviewer for the suggestion, and while we agree that including larger polyphenol would be interesting the analytical complexity of using the larger polyphenols would make monitoring the kinetics difficult. This complexity is why we decided to focus on small phenols to use as model compounds.

Comment 16: It can’t be said that “gallic acid was a slower reaction”, please rephrase, e.g. “enzymatic degradation of gallic acid by MnP was found to be slower compared to pyrogallol as substrate”.

Response 16: We would like to thank the reviewer for the thoughtful review of our work. This sentence is referring to the abiotic degradation in our previous work. As such, rephrasing the sentence would not be appropriate.

Comment 17:  Abiotic degradation was not part of this investigation, so it can’t be said that “it was demonstrated in this study”. Maybe, if this was better explained at the beginning, to define what is considered biotic degradation, and what abiotic degradation, what is the pattern of abiotic degradation (from previous study) and the one confirmed in this study.

Response 17: See response to the next comment.

Comment 18: It seems that the results are overestimated, or that the conclusions are too generalised (PSM without the carboxylic group) with the assertion that “it was demonstrated that PSM that do not have a carboxylic acid functional group are likely to form a quinone and contribute to the formation of SOM, and it should be removed.

Response 18: Again, we would like to thank the reviewer for carefully reviewing the manuscript. The authors believe that an error in our editing contributed to this confusion. There was a word missing in the referenced sentence. We have fixed that error and have clarified the conclusions. Lines 293-313

Reviewer 2 Report

Comments and Suggestions for Authors

  • The title can be changed with “Investigation of Manganese Peroxidase Kinetics with Simple Phenolic Compounds” to be more descriptive.
  • The abstract summarizes the main findings but does not clearly emphasize the study's significance and contribution to the existing literature. Adding a brief statement on its relevance would enhance clarity.
  • The source information for chemicals is listed, but details on their purity or grade (e.g., analytical, reagent) are not provided. Including this information would improve reproducibility.
  • The description of the sodium acetate buffer is somewhat unclear.
  • The enzyme reaction conditions were adapted from Xu et al. [17], but no explanation is provided for why this particular study and its conditions were chosen.
  • The incubation time for enzyme kinetics was set to 8 minutes, but no rationale is given for this choice. It should be clarified whether this duration was selected to capture the initial reaction rates effectively.
  • While kinetic data analysis is mentioned, details on error estimation (e.g., standard error or confidence intervals) and statistical significance thresholds are missing.
  • The conclusion restates the findings but lacks a broader discussion of their significance and potential implications for future research or practical applications.

Author Response

Summary. We would like to thank the reviewer for their careful review of our work. We have addressed the concerns of the reviewer below. 

Comment 1: The title can be changed with “Investigation of Manganese Peroxidase Kinetics with Simple Phenolic Compounds” to be more descriptive.

Response 1: Thank you for the suggestion. We have revised the title to: Kinetics of Manganese Peroxidase Using Simple Phenols as Substrates

Comment 2: The abstract summarizes the main findings but does not clearly emphasize the study's significance and contribution to the existing literature. Adding a brief statement on its relevance would enhance clarity.

Response 2: We would like to thank the reviewer for this suggestion. We have added a sentence to lines 14-17

Comment 3: The source information for chemicals is listed, but details on their purity or grade (e.g., analytical, reagent) are not provided. Including this information would improve reproducibility.

Response 3: Thank you for the suggestion. The stated purity from the manufacturer has been added to line 133-135.

Comment 4: The description of the sodium acetate buffer is somewhat unclear.

Response 4: We have clarified the buffer line 131-133

Comment 5: The enzyme reaction conditions were adapted from Xu et al. [17], but no explanation is provided for why this particular study and its conditions were chosen.

Response 5: We have added a sentence on why we used the cited study as the starting point for our conditions. Line 137-138

Comment 6: The incubation time for enzyme kinetics was set to 8 minutes, but no rationale is given for this choice. It should be clarified whether this duration was selected to capture the initial reaction rates effectively.

Response 6: We present the rationale in the results of 3.1 (lines 197-209)

Comment 7: While kinetic data analysis is mentioned, details on error estimation (e.g., standard error or confidence intervals) and statistical significance thresholds are missing.

Response 7: Thank you for the comment. We have added the error bar information in the graphs and indicated how we determined the differences in line 177-181

Comment 8: The conclusion restates the findings but lacks a broader discussion of their significance and potential implications for future research or practical applications.

Response 8: We have reworded the conclusion to incorporate this comment. Please refer to the red track changes.

Reviewer 3 Report

Comments and Suggestions for Authors

The authors discussed secondary metabolites, a diverse compound class that includes slight phenols and large polymeric structures like tannins. Minor compounds like phenolic acids can serve as substrates for soil microbes and enzymes. In this study, we monitored the activity of a redox-active soil enzyme called manganese peroxidase (MnP) concerning three slight phenols: pyrogallol, gallic acid, and benzoic acid. Based on the kinetic parameters we determined, both pyrogallol and gallic acid act as substrates for MnP, although they yield different products and exhibit distinct kinetics. Specifically, pyrogallol reacts more quickly and produces a more stable quinone than gallic acid. In contrast, benzoic acid is not a substrate for MnP.   Therefore, this manuscript is well-written and addresses an important topic: secondary metabolites. It can be accepted following a review that includes enhancing the introduction with more recent research, thoroughly examining the results, and discussing them in the context of previous findings. Additionally, a section should be added to the conclusion that explains the significance of this study, its potential benefits, and future directions for serving both the environment and society.  

Comments on the Quality of English Language

The English language used in the research could be enhanced to express the findings more clearly.

Author Response

Summary: We would like to thank the reviewer for their kind words about our work. We have address the concerns about the abstract, introduction, and discussion. 

Comment 1: The authors discussed secondary metabolites, a diverse compound class that includes slight phenols and large polymeric structures like tannins. Minor compounds like phenolic acids can serve as substrates for soil microbes and enzymes. In this study, we monitored the activity of a redox-active soil enzyme called manganese peroxidase (MnP) concerning three slight phenols: pyrogallol, gallic acid, and benzoic acid. Based on the kinetic parameters we determined, both pyrogallol and gallic acid act as substrates for MnP, although they yield different products and exhibit distinct kinetics. Specifically, pyrogallol reacts more quickly and produces a more stable quinone than gallic acid. In contrast, benzoic acid is not a substrate for MnP.  

Therefore, this manuscript is well-written and addresses an important topic: secondary metabolites. It can be accepted following a review that includes enhancing the introduction with more recent research, thoroughly examining the results, and discussing them in the context of previous findings. Additionally, a section should be added to the conclusion that explains the significance of this study, its potential benefits, and future directions for serving both the environment and society.  

Response 1: We would like to thank this reviewer for their reviewer. We have addressed the comment in the introduction, discussion, and conclusion. Please refer to the red track changes.

Reviewer 4 Report

Comments and Suggestions for Authors
  1. Dear colleagues! The results you have obtained are very interesting. However, there is a certain nuance. Benzoic acid, strictly speaking, is not a phenolic compound. Therefore, it should be considered as a model of an aromatic compound that does not contain a hydroxyl group. Therefore, it would be correct to change the title of the article and to make appropriate amendments to introduction, experimental section discussion and conclusions.
  2. The lack of contribution from the carboxyl group is also a debatable issue. Pyrogallol and gallic acid contain an equal number of hydroxyl groups, but the kinetic curves differ significantly. Therefore, in my opinion, the last sentence of Section 4 should be corrected. Perhaps, changes should be made to the conclusions.
  3. If the authors are going to continue research in this direction, it would make sense to test, for example, salicylic, caffeic, and ferulic acids, as well as phenols containing 2 hydroxyl groups.

Author Response

Summary: We would like to thank the reviewer for taking the time to reviewer our manuscript. We have addressed the comments below. 

Comment 1: Dear colleagues! The results you have obtained are very interesting. However, there is a certain nuance. Benzoic acid, strictly speaking, is not a phenolic compound. Therefore, it should be considered as a model of an aromatic compound that does not contain a hydroxyl group. Therefore, it would be correct to change the title of the article and to make appropriate amendments to introduction, experimental section discussion and conclusions.

Response 1: We have amended the manuscript to reflect this change.

Comment 2. The lack of contribution from the carboxyl group is also a debatable issue. Pyrogallol and gallic acid contain an equal number of hydroxyl groups, but the kinetic curves differ significantly. Therefore, in my opinion, the last sentence of Section 4 should be corrected. Perhaps, changes should be made to the conclusions.

Response 2: Thank you for your thoughtful comments. We have edited the end of the discussion to reflect this, lines 280-287.

Comment 3: If the authors are going to continue research in this direction, it would make sense to test, for example, salicylic, caffeic, and ferulic acids, as well as phenols containing 2 hydroxyl groups.

Response 3: We would like to thank the reviewer for this suggestion.

Round 2

Reviewer 1 Report

Comments and Suggestions for Authors

Reviewer comment to Authors' Responses (V2)  to Reviewer's Comments (V1)

I appreciate the consideration of comments to the first version of the manuscript. All comments were addressed, and I agree with almost all of them. A few details are still doubtful, so I am kindly asking for consideration of the next few comments.

  1. Reviewer’s Reply to Authors’ Response no.1.

Considering the text from Marchiosi et al., 2020 (doi:10.1007/s11101-020-09689-2) which describes the group of compounds having a phenolic structure with respect to other classes of plant’s secondary matabolites:

“The diversity of secondary compounds in the plant kingdom is huge. About 200,000 compounds are known, which are grouped into amines, nonprotein amino acids, peptides, alkaloids, glucosinolates, cyanogenic glucosides, organic acids, terpenoids, quinones, polyacetylenes, and phenolics. The group of phenolic compounds consists of polyphenols, oligophenols and monophenols or simple phenolic compounds such as benzoic and cinnamic acids and their hydroxylated derivatives. Among the thousands of compounds present in ecological interactions, simple phenolic acids are the most abundant in soils, and many are described as allelochemicals.”

it can be said that the change made to the manuscript isn’t correct because the meaning of the change in manuscript is that the phenols are the only group of plant secondary metabolites, which is not true. Please revise in abstract and introduction, e.g.:

Secondary metabolites encompass diverse groups of compounds, among which the group of compounds with a phenolic structure ranging from small phenols to larger polyphenols such as lignin and tannins.

  1. Regarding the section Data analysis:

Please add the full name of the software used: GraphPad software, Prism version 10.2.0, because it seems you omitted part of information from the version 1 of the manuscript.

The sentence in the former version:

“Differences between values was determined using a sum of least squares F test.”

was substituted by:

“Statistically significant differences between the Km and Vmax means were determined using a sum of squares F test.”

It seems I added confusion with the comment, but the text as it was in V1, was not clear enough. A precise description of the performed statistic is recommended. So actually, with this software you were able to create a model for enzyme kinetics, obtain the kinetic parameters Vmax and Km for each treatment, and based on F-test you tested for significance the models describing the kinetics?  Please check if this is true and correct accordingly.

  1. Figure 2. isn’t named correctly. It contains the description of materials and methods which are already described in appropriate text sections. The name should be e.g.: Figure 2. Manganese Peroxidase reaction rates with three different substrates.

  1. Please correct “triplicated” with “triplicate” in Figure 2 and Figure 3.

Regarding Authors’ Response no.12., I apologize to the authors, it was truly a misunderstanding on my part.

Kind regards

Author Response

Summary: The authors would like to thank the reviewer for their thorough and comprehensive evaluation of our work. We believe all of the comments have been fully addressed. 

Comment 1: 

  1. Reviewer’s Reply to Authors’ Response no.1.

Considering the text from Marchiosi et al., 2020 (doi:10.1007/s11101-020-09689-2) which describes the group of compounds having a phenolic structure with respect to other classes of plant’s secondary matabolites:

“The diversity of secondary compounds in the plant kingdom is huge. About 200,000 compounds are known, which are grouped into amines, nonprotein amino acids, peptides, alkaloids, glucosinolates, cyanogenic glucosides, organic acids, terpenoids, quinones, polyacetylenes, and phenolics. The group of phenolic compounds consists of polyphenols, oligophenols and monophenols or simple phenolic compounds such as benzoic and cinnamic acids and their hydroxylated derivatives. Among the thousands of compounds present in ecological interactions, simple phenolic acids are the most abundant in soils, and many are described as allelochemicals.”

it can be said that the change made to the manuscript isn’t correct because the meaning of the change in manuscript is that the phenols are the only group of plant secondary metabolites, which is not true. Please revise in abstract and introduction, e.g.:

Secondary metabolites encompass diverse groups of compounds, among which the group of compounds with a phenolic structure ranging from small phenols to larger polyphenols such as lignin and tannins.

Response 1: We have reworded the abstract to the following: 

Secondary metabolites encompass diverse groups of compounds, one such group is phenolics that include small phenols to larger polyphenols such as lignin and tannins. 

We reworded the introduction to:

Secondary metabolites are a diverse class of compounds. One subset of PSM are phenolics that encompass compounds ranging from simple phenolic acids to large polymeric structures such as lignin and tannins 

Comment 2: 

  1. Regarding the section Data analysis:

Please add the full name of the software used: GraphPad software, Prism version 10.2.0, because it seems you omitted part of information from the version 1 of the manuscript.

The sentence in the former version:

“Differences between values was determined using a sum of least squares F test.”

was substituted by:

“Statistically significant differences between the Km and Vmax means were determined using a sum of squares F test.”

It seems I added confusion with the comment, but the text as it was in V1, was not clear enough. A precise description of the performed statistic is recommended. So actually, with this software you were able to create a model for enzyme kinetics, obtain the kinetic parameters Vmax and Km for each treatment, and based on F-test you tested for significance the models describing the kinetics?  Please check if this is true and correct accordingly.

Response 2: We have changed that portion to:

Graph Pad Prism software version 10.2.0 (392) was used to determine the kinetic parameters (Km and Vmax) using a least squares regression of the data. A sum of squares F-test was used to determine if the best-fit values of each parameter (Km and Vmax) were statistically different.

Comment 3: Figure 2. isn’t named correctly. It contains the description of materials and methods which are already described in appropriate text sections. The name should be e.g.: Figure 2. Manganese Peroxidase reaction rates with three different substrates.

Response 3: We had named the figure exactly as the reviewer indicated. 

Comment 4: Please correct “triplicated” with “triplicate” in Figure 2 and Figure 3.

Response to comment 4: We have made the correction to both figures.